# Robust cortical encoding of 3D tongue shape during feeding in macaques

Jeffrey D. Laurence-Chasen[1] ✉, Callum F. Ross[1], Fritzie I. Arce-McShane [2,3,5] & Nicholas G. Hatsopoulos [1,4,5]

Dexterous tongue deformation underlies eating, drinking, and speaking. The orofacial sensorimotor cortex has been implicated in the control of coordinated tongue kinematics, but little is known about how the brain encodes—and ultimately drives—the tongue's 3D, soft-body deformation. Here we combine a biplanar x-ray video technology, multi-electrode cortical recordings, and machine-learning-based decoding to explore the cortical representation of lingual deformation. We trained long short-term memory (LSTM) neural networks to decode various aspects of intraoral tongue deformation from cortical activity during feeding in male Rhesus monkeys. We show that both lingual movements and complex lingual shapes across a range of feeding behaviors could be decoded with high accuracy, and that the distribution of deformation-related information across cortical regions was consistent with previous studies of the arm and hand.

The sensorimotor cortex encodes various characteristics of the musculoskeletal movements that make up every-day behaviors such as walking, reaching, and grasping[1,2]. But not all coordinated motor actions involve bones moving about joints; the tongue is a muscular hydrostat unconstrained by rigid internal structure[3] which performs rapid, complex deformations during eating, drinking, and speaking[4]. Orofacial sensorimotor cortex (OSMCx) is known to be involved in the control of tongue movements[5–11], but the extent to which 3D tongue shape is encoded by the sensorimotor cortex has not previously been evaluated.

Intraoral tongue deformation (shape change) is notoriously difficult to measure[4]; the tongue is almost entirely obscured from view by lips, cheeks, teeth, and jaws. Consequently, prior studies of the cortical control of tongue kinematics have been restricted to measuring inferred[6], extraoral[12], or 2D[8,13] tongue motion only. In this study, we use biplanar videoradiography and deep neural networks to measure and decode a rich set of 3D intraoral tongue kinematics. Understanding the cortical representation and control mechanisms of such soft-body kinematics is a central goal of orofacial neuromechanics and soft robotics[14,15], and is essential for future development of rehabilitative technologies.

## Results

### Quantifying intraoral tongue kinematics

To measure simultaneous intraoral tongue kinematics and related cortical activity we used a combination of XROMM (X-ray reconstruction of moving morphology[16]) and intracortical microelectrode array recording, respectively (Fig. 1a–e). Biplanar videoradiography and the XROMM workflow enable high-resolution measurement of intraoral tongue kinematics and have recently yielded new insight into 3D tongue motions during feeding;[17–20] the tongue deforms in complex and varied ways as it transports food to the molars (stage 1 transport), manipulates it into a bolus during mastication, moves it into the oropharynx (stage 2 transport), and, ultimately, squeezes it into the esophagus during swallowing[4]. We imaged the motion of a constellation of 7 implanted tongue markers (1.0 mm diameter tantalum beads; Fig. 1a, c, d) at 200 Hz in two Rhesus macaque monkeys (Ry and Ye) feeding on grapes. Using a standard marker-based XROMM

[1]Department of Organismal Biology and Anatomy, The University of Chicago, 1027 E 57th Street, Chicago, IL 60637, USA. [2]Department of Oral Health Sciences, School of Dentistry, University of Washington, 1959 NE Pacific StreetBox #357475 Seattle, WA 98195-7475, USA. [3]Graduate Program in Neuroscience, University of Washington, 1959 NE Pacific St., Seattle, WA 98195-7475, USA. [4]Program in Computational Neuroscience, The University of Chicago, 5812 South Ellis Avenue, Chicago, IL 60637, USA. [5]These authors contributed equally: Fritzie I. Arce-McShane, Nicholas G. Hatsopoulos. ✉e-mail: jd.laurencechasen@gmail.com

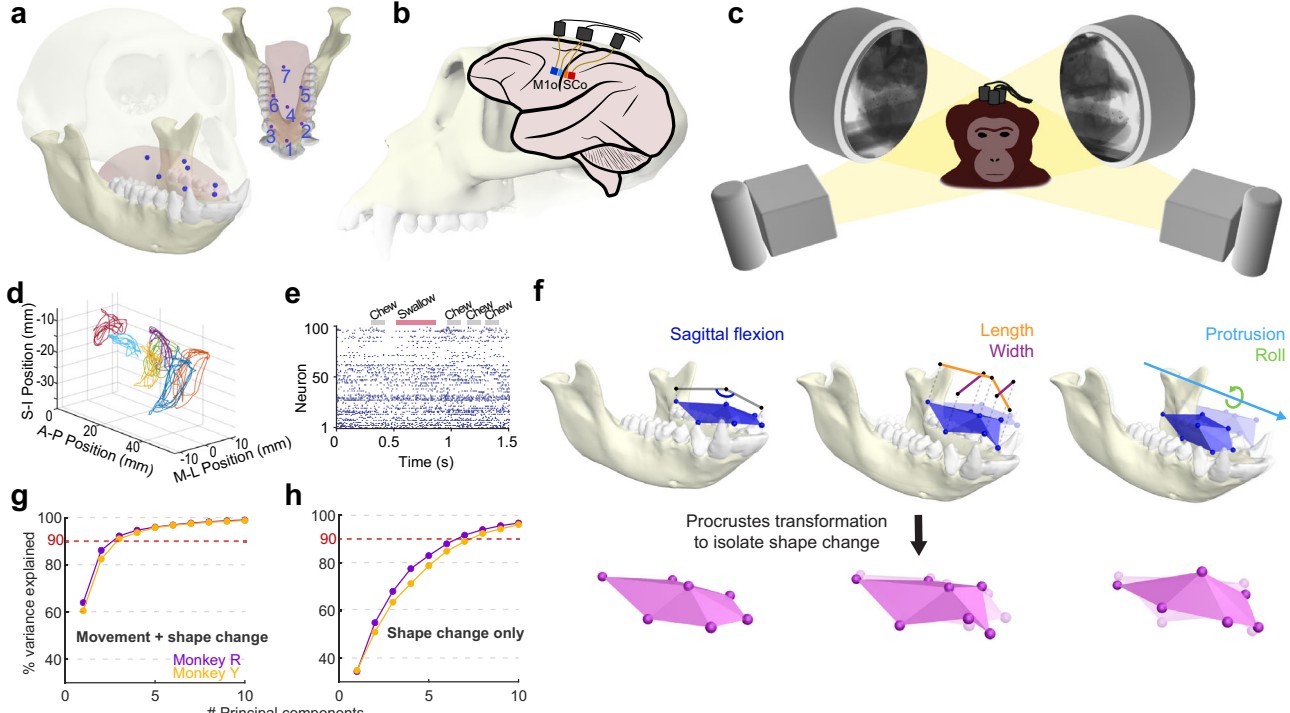

**Fig. 1 | Quantifying intraoral tongue deformation and related cortical responses. a** A constellation of 7 radio-opaque markers (blue spheres) was implanted into the tongue body to capture whole-tongue kinematics. **b** Multielectrode arrays were implanted in the orofacial region of the primary motor cortex (M1o; dark blue, Utah array; light blue, floating microelectrode array) and somatosensory cortex (SCo; red, Utah array; orange, floating microelectrode array). **c, d** While the subjects fed on grapes, biplanar videoradiography (**c**) recorded the 2D, intraoral motion of the markers, from which 3D trajectories (**d**) were computed. In **d**, different colors represent different tongue markers, S-I, superoinferior; A-P, anteroposterior; M-L, mediolateral, all relative to the cranium.

**e** Spike raster of neural data (100 representative neurons from M1o) collected synchronously with the kinematics shown in **d. f** Top, digital renders of tongue and mandible posture at three timepoints in a chewing cycle and computed kinematic variables. Bottom, a constrained Procrustes superimposition was performed to remove translational, rotational, and scale changes in marker positions, leaving only shape change. **g** Percent variance explained by first 10 components of a principal component analysis on marker positions (relative to the cranium) across all trials. **h** Same as **g**, but computed on marker positions in Procrustes shape space. See "Methods: XROMM data processing" for details on image generation.

workflow[17,19], we reconstructed the 3D positions of the 7 tongue markers relative to the cranium. A principal component analysis (PCA) of the XYZ marker positions across all trials found that the first 3 components accounted for 90% of the total tongue kinematic variance (Fig. 1g), but only 70% of the total tongue shape variance (Fig. 1h; Supplementary Movie 1).

Extracting kinematic variables that generalize across subjects in the absence of rigid bones and joints is a fundamental challenge in lingual biomechanics[19,20]. Here we chose to use two approaches in an attempt to balance considerations of generalizability and dimensionality (i.e., capturing the complexity of tongue deformation): a biomechanics-based approach using standard tongue kinematic variables[19] and a Procrustes-based approach using the principal components of tongue shape.

## Decoding tongue movement

We first used an LSTM[21] network to independently predict tongue movement variables from the responses of a population ($n = 100$) of orofacial primary motor cortex (M1o) neurons. The LSTM architecture was chosen for its demonstrated ability to achieve high decoding performance without assuming linearity[22]. Neuronal activity was recorded with Utah arrays and floating microelectrode arrays (FMAs; Fig. 1b, e). The positions of implanted tongue markers themselves lack inherent biomechanical significance, so we calculated a set of standard tongue kinematic metrics from the XYZ marker positions (Fig. 1f, top): sagittal flexion, roll, protrusion, as well as regional lengths and widths. Notably, tongue roll is a mediolateral, asymmetrical motion not

captured by 2D lateral imaging, and has received relatively little attention despite its key role in feeding[19].

Using $R^2$ (in the "fraction of variance accounted for" sense[23]) as our performance metric, we found that all variables were accurately decoded on cross-validated data from M1o activity across the range of functional stages of feeding (Fig. 2). In both monkeys, tongue roll was decoded most accurately. The range of mean decoding accuracy of all variables was 0.43–0.85 (Fig. 2b), well within benchmarks for the arm and hand[24,25].

Due to the cyclic nature of mastication, there are abundant correlations between tongue kinematic variables and those of the jaw[19] (Supplementary Fig. 1). To ensure that our decoding performance was not simply a consequence of the decoder learning and exploiting that correlational structure, we systematically investigated the relationship between tongue-jaw correlation and decoding accuracy (Supplementary Fig. 4). Through iterative sampling of sub-regions of the test trials, we found that correlation of tongue kinematic variables with mandibular motion does not account for decoding accuracy. Even at times where tongue motion was completely un-correlated with the jaw, decoding accuracy could be quite high.

## Decoding tongue shape

We next examined the extent to which tongue shape alone could be decoded from M1o. To that end, we performed a Procrustes superimposition to remove translational, rotational, and scale changes in tongue posture (Fig. 1f, bottom). The Procrustes superimposition yielded a new set of marker coordinates which contained only tongue

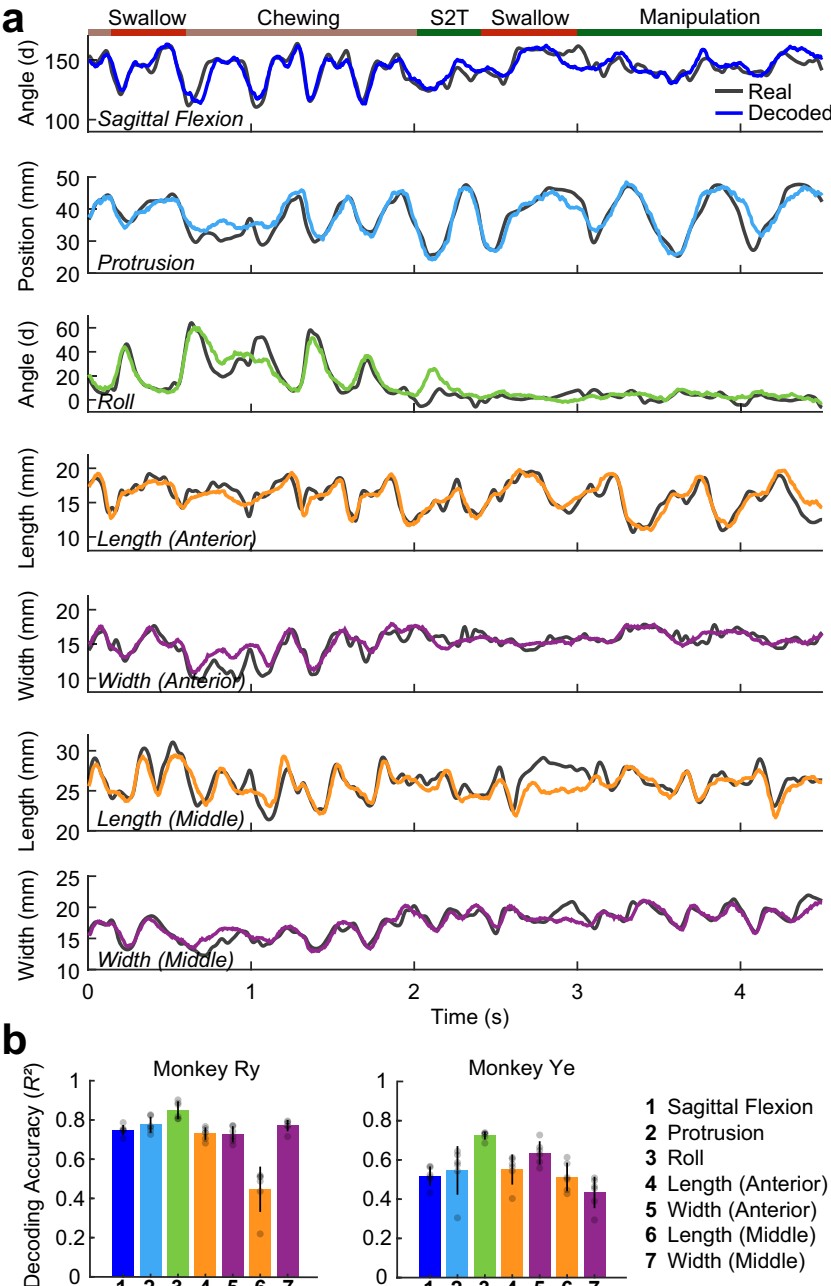

**Fig. 2 | Decoding tongue kinematics from motor cortex. a** Decoded (colored lines) and ground-truth (grey lines) kinematics of 7 standard tongue movement variables, each independently decoded for a subsection of a representative test trial. The colored bar at the top of the figure indicates the behavior being performed. S2T denotes stage 2 transport. (See Supplemental Methods for detail on gape cycle types.) **b** Mean decoding accuracy ($R^2$), by variable, for both subjects. Each bar represents the mean decoding accuracy for that variable across the cross-validation folds ($n = 6$). Error bars represent ±1 standard deviation.

shape information. Changes in tongue shape during feeding were complex; 7 principal components were required to account for the majority (>90%) of total shape variance (Fig. 1h), In order to preserve this complexity, we used the scores of those first 7 PCs (in order of % variance explained) as "complex deformation" variables to be decoded. This approach resembles one used in previous studies which decoded principal components as a means of investigating joint and muscle synergies[24,26].

We found that multiple PCs of tongue shape could be decoded with high accuracy (Fig. 3 and Supplementary Movie 2). Notably, the first two shape PCs were correlated with sagittal flexion and roll, important elements of tongue deformation during feeding[19]

(Supplementary Figs. 1 and 2). Thus, we were unsurprised at their high decoding accuracy. However, PCs 3–7 represented more complex, compound shape changes that are not readily attributable to a single standard kinematic variable. The decoding accuracy of these smaller-variance PCs was lower, but we were still able to reconstruct whole-tongue shape change with sub-millimeter accuracy from independently decoded shape PCs (Supplementary Fig. 3). While the negligible decrease in reconstruction error with the addition of PCs 5–7 suggests that the decoder may not be capturing neural variance that is relevant to the most subtle aspects of tongue shape, decoding was possible even at periods where the correlation of shape PCs with kinematics variables was low (Supplementary Fig. 9).

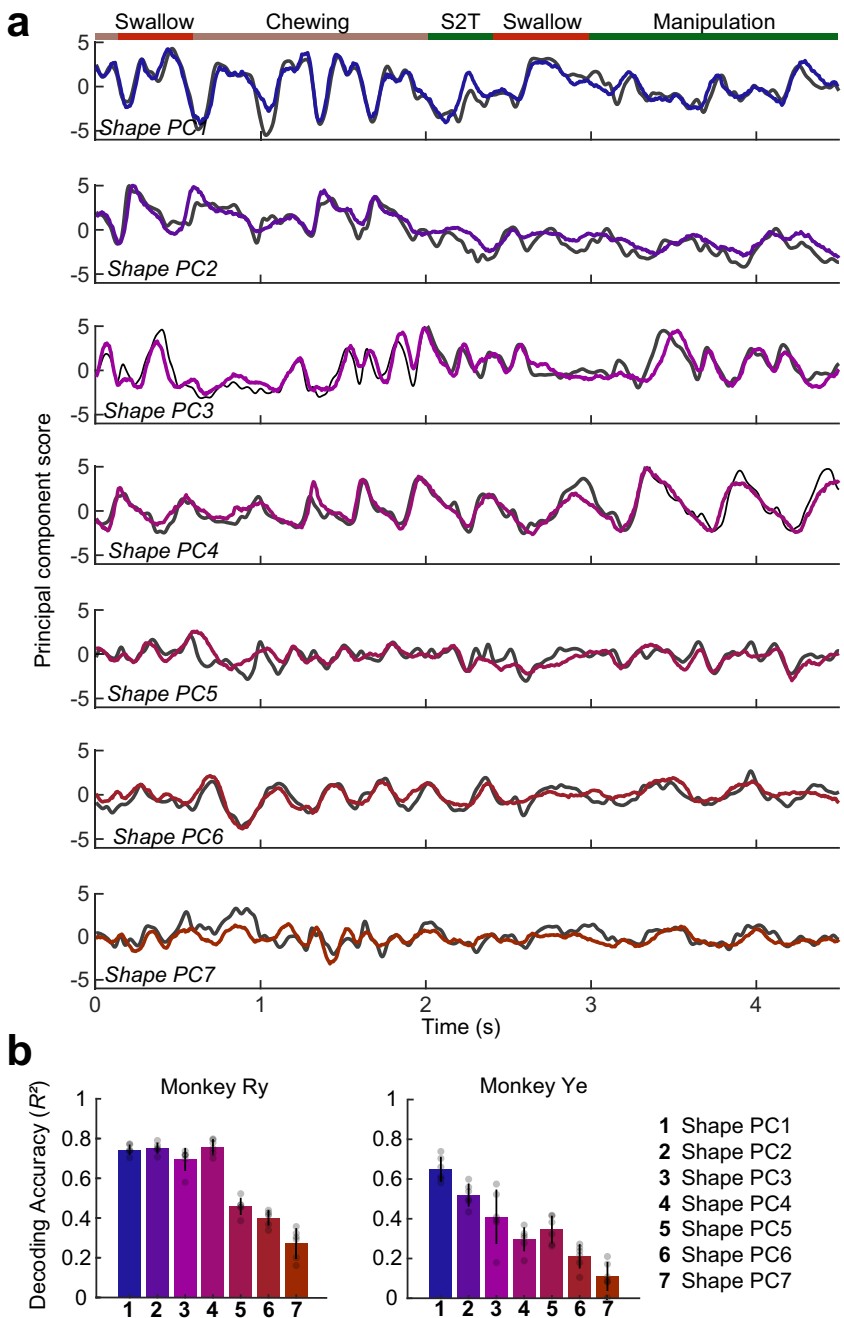

**Fig. 3 | Decoding tongue shape from motor cortex. a** Decoded (colored lines) and ground-truth (grey lines) kinematics of the first 7 principal components of tongue shape (PCA of Procrustes-transformed marker positions), each independently decoded for a subsection of a representative test trial. The colored bar at the top of the figure indicates the behavior being performed. S2T denotes stage 2 transport (See Supplementalary Methods for detail on gape cycle types.) **b** Mean decoding accuracy ($R^2$), by variable, for both subjects. Each bar represents the mean decoding accuracy for that variable across the cross-validation folds ($n = 6$). Error bars represent ±1 standard deviation.

## Decoding of tongue-related information differs between M1o and SCo

In the limb, high-accuracy decoding can be achieved from small populations of both primary motor cortex (M1) neurons and somatosensory cortex (SC) neurons. To test whether this is also true of the tongue from OSMCx, we trained decoders with the same number of neurons ($n = 55$) from each cortical area on identical kinematic datasets. We found that, in both monkeys, M1o decoding accuracy was significantly better than that of SCo (Fig. 4a; $P < 0.0001$, Wilcoxon Signed Rank Test).

After determining that M1o populations contain more tongue-related information, we next assessed the extent to which that information was distributed across populations of M1o neurons. We varied the number of neurons used as decoder input from 1 to 100, randomly drawing sub-populations at each ensemble size (Fig. 4b). We found that decoding accuracy for both variable types began to plateau at approximately 25-35 neurons, but continued to increase, albeit at a slower rate, as the ensemble grew to 100 neurons. These results are remarkably consistent with previous studies in the arm and hand, which, although using completely different decoders, found a similar

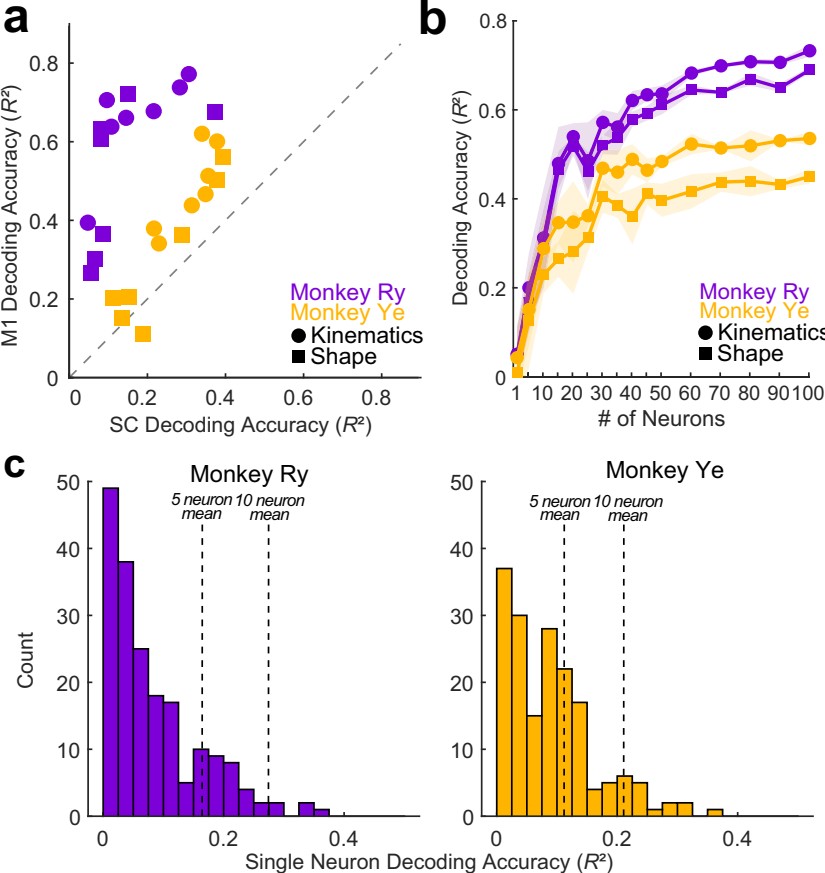

**Fig. 4 | Region and ensemble size effects on decoding accuracy. a** Comparison of decoding accuracy from M1o and SCo populations of equal size ($n = 55$ neurons) for all variables. Each shape represents the mean decoding accuracy for a single variable across 6 folds of cross-validation. Notably, for all but one variable for both subjects, M1o decoding was better than SCo. Circles are kinematic variables and squares are shape variables. **b** Decoding accuracy (mean of all variables) as a function of M1o neuronal ensemble size. Circles are kinematic variables and squares are shape variables. Shape variable values were weighted based on the given PC's fraction variance explained when averaging. Shaded error region represents ±1 standard deviation. **c** Distribution of decoding accuracy of a subsample ($n = 40$ per monkey) of high-performing single M1o neurons. Input to histogram was the decoding accuracy of individual variables (pooled from all 40 neurons) whose $R^2$ was positive. Thus the distribution represents the right-tail of the performance distribution. Vertical dashed lines indicate the mean accuracy (across all variables) for ensembles of 5 and 10 M1o neurons.

ensemble size-performance relationship[25,27]. Furthermore, in Fig. 4b, while the shape of the accuracy versus ensemble size curves are relatively similar across the two monkeys, there is a clear vertical shift (i.e., difference in decoding performance) between monkey Ry and monkey Ye. The similarity of curve shape indicates a similar distribution of shape-related information within the two populations of sampled neurons. This similarity emerges despite the likelihood of slight interindividual differences in array placements in the cortex that may explain differences in overall decoding accuracy.

We next assessed the extent to which tongue-related information was present in the firing of individual cortical neurons. For a subset of decoders trained with single-neuron inputs, we examined the distribution of average decoding accuracy for individual variables. Single neuron decoding accuracy varied widely, with the majority of single neurons failing to achieve high-accuracy decoding for any variables (Fig. 4c). However, both monkeys had a small subset of neurons whose decoding performance was higher than the average performance of decoders trained with 5 and even 10 times as many cells (Fig. 4c, right tail of distribution). A permutation test of single-neuron decoding accuracy values after shuffling the neural and kinematic data suggested that the likelihood of observing these results by chance is extremely low ($p < 0.0001$; 10,000 permutations; see Methods). Further inspection of the decoding performance of select neurons illustrated instances of neuronal "tuning" to different tongue shape

parameters (Supplementary Fig. 10). This unequal (at the level of individual neurons) distribution of movement information is well documented in both the upper limb region as well as the orofacial region of M1[27].

Taken together, we infer that exact array location may be a major factor in absolute decoding performance, but that the general distribution of movement- and shape- related information within subpopulations of M1 neurons is relatively consistent across different functional areas (i.e., orofacial and upper limb)[28].

## Discussion

The tongue is a muscular hydrostat lacking joints and capable of complex, nonlinear deformation[3]. Using deep neural networks to decode tongue movement and deformation from OSMcx during feeding, we found that information about 3D tongue shape is present and accessible in M1o neuronal ensembles of various sizes. Our results build upon previous studies which demonstrated that tongue protrusion direction and tongue tip position can be decoded from cortex with the same methods used commonly in the upper limb[5,8,29]. Specifically, various decoding algorithms have previously been used to successfully predict both hand direction[27,30] and 3D posture—in the form of finger-joint angles[24,31,32] from non-human primate M1.

Though the hand and tongue are anatomically disparate (the tongue is a muscular hydrostat with no internal joints[3]) the two

effectors exhibit striking functional similarities[33]. Both rapidly change their 3D posture to deftly control food and other objects[4,20,34], and dexterity in both is enabled by rich mechanosensory innervation that provides a wealth of ongoing feedback to the brainstem and sensorimotor cortex[35–37]. Moreover, the relative complexity of tongue shape and hand posture appear to be similar[38]. We found that on the whole the cortical representation of tongue shape and movement is consistent with this functional analogy. Individual M1o neurons contained a variable amount of tongue movement-related information (Fig. 4c), and decoding performance began to plateau at approximately 25–35 cells (Fig. 4b). We achieved consistently higher decoding accuracy from populations of M1o as compared to SCo (Fig. 4a). This result was surprising, as the tongue and oral cavity are richly innervated with various types of mechanoreceptors, and various studies have demonstrated the complex receptive field structure of cortical orofacial sensory neurons[39,40]. Overall, the fact that decoding is comparable for tongue and limbs is not a priori expected. An alternate outcome would have been that sensorimotor cortex decodes gross movements of the tongue such as flexion and protrusion, but not detailed tongue shape.

The nature of the experimental behavior is an important factor in generalizing findings beyond the specifics of a single study. Motor neuroscience experiments have typically involved extensively training animals to perform discrete movements such as reaching in different directions or grasping various objects (e.g., refs. 24,41,42). Our task differed from this structure in that it was a cyclic, naturalistic behavior that involved no training. Animals initiated and completed feeding sequences freely. Additionally, feeding is also punctuated by discrete events—swallows—that are voluntarily initiated but rich in reflexive components. We believe all of these factors strengthen the present study; though repetitive, the behavior was compound cyclic-discrete in nature, and the different stages of feeding (ingestion, chew, swallow) elicited a range of kinematics. Importantly, it is likely that variation in tongue kinematics within a feeding sequence is substantially greater than variation between sequences of similar and even different food types[19,43].

Without joints, the tongue's theoretical dimensionality, or number of degrees of freedom, is large. However, there are a finite number of muscles (-16) and motor units within the tongue[44]; and realistically constrained computational models have yielded impressive results[45,46]. Our results demonstrate that information about complex tongue shapes and movements can be summarized in relatively few (<10) dimensions (Fig. 1h), and much of that information is represented in M1o activity. There were inter-individual differences in aspects of tongue shape that the shape PCs captured (Supplementary Fig. 2), which may simply be a consequence of the underlying mathematics of PCA when components explain similar amounts of variance. However, there were also clear patterns of similarity between the shape PCs, in that, in both subjects, shape PC1 captured elements of sagittal flexion and shape PC3 captured posterior tongue elevation likely related to swallowing. Further studies should investigate the cortical encoding and decoding of the specific behaviors that feeding comprises (i.e., swallowing, transport).

Much is still unknown about the fundamental neuromechanical mechanisms of lingual control. In particular, feeding is semi-automatic and is typically assumed to involve structures outside of cortex. The fact that M1 and SC can decode details of tongue shape suggest at a minimum that sensorimotor cortex is informed of the detailed kinematics and shape of the tongue during feeding. It may indicate that the cortex is involved in driving behavior in a soft-tissue effector.

Our results have significant implications for the development of lingual neuroprostheses. Currently, individuals who experience total loss of tongue function or full glossectomy have few options for regaining tongue function[47]. Mandibular and palatal prostheses exist, but do not offer any active aid in speaking or swallowing[48]. The finding that 3D tongue posture can be accurately decoded from the sensorimotor cortex opens up a new avenue for potential brain computer interface-based prostheses for restoring orolingual function and communication[49]. As the field of soft-robotics continues to flourish[15], the reality of such a device becomes increasingly likely.

## Methods

### Animals and surgery
We recorded kinematics and cortical neural activity from two adult male rhesus macaques (monkeys Ry and Ye; *Macaca mulatta*, 9–10 kg). Monkeys received full-time care from husbandry and veterinary staff, and all protocols were approved by the University of Chicago Animal Care and Use Committee and complied with the National Institutes of Health Guide for the Care and Use of Laboratory Animals. Surgical procedures consisted of the implantation of radiopaque beads for marker-based XROMM[16] and the implantation of intracortical microelectrode arrays for the recording of neural activity[10,50].

In the marker implantation surgery, an angiocatheter and stylus were used to insert 15 radiopaque beads (tantalum, 1 mm diameter) into the tongue at various positions and depths following previously described methods[17]. In brief, beads were implanted at 15–25 mm intervals down the anterior–posterior axis of the tongue, with 2–3 beads in each layer (distributed across the coronal plane), except the tongue tip, which had one bead. For the analysis in this study, a subset (7) of the middle and anterior tongue beads, each at approximately 3-5 mm depth, were used (Fig. 1a). These beads were selected for their consistent locations between the two individuals and their uniform distribution across the anterior and middle tongue. Additional beads were implanted into the cranium and mandible (4 per bone) using a standard, drill-based technique[16].

In the array surgery, each monkey was implanted with two Utah arrays (Blackrock Microsystems, Inc., Salt Lake City, UT), and two floating microelectrode arrays (FMA, Microprobes for Life Science, Gaithersburg, MD) (Fig. 1b). Prior to the surgeries, individual-specific surgical plans were established through a multi-modal approach. For each monkey, 3D mesh models of the cranium and brain were generated from CT and MRI scans, respectively, using 3D Slicer[51] (www.slicer.org). The models were then manually registered (aligned) in Maya 2020 (Autodesk, San Rafael, CA, U.S.A), and the approximate location of the orofacial sensorimotor cortex was identified as rostral to the tip of the intraparietal sulcus on the brain model. The corresponding superficial location was then identified on the skull model and the coordinates of that location relative to bregma were recorded and used to inform intra-operative craniotomy location. After the craniotomy, surface electrical simulation (and its evoked movements) was used to identify the borders of the orofacial sensorimotor cortex. Utah arrays (96 electrodes; M1 electrode length: 1.5 mm; SC electrode length: 1.0 mm) were implanted into orofacial region of rostral M1 and areas 1/2 of the somatosensory cortex. Floating microelectrode arrays (32 electrodes; M1 electrode length: 3–4.5 mm; SC electrode length: 4.0–8.7 mm) were implanted into caudal M1 and area 3a/3b[9,10]. Array arrangement can be seen in Fig. 1b and Supplementary Fig. 8. A post-operative CT scan was taken, and 3D models of the array and electrodes were generated, registered, and combined with the pre-existing cranium and brain models, to confirm correct electrode placement.

### Behavioral task and dataset composition
Subjects received and consumed food items while head-fixed and seated in a standard primate chair in the University of Chicago XROMM Facility. Experimental food comprised half grapes of equal size presented directly to the monkey's mouth via a long stylus. Trials began with the depression of the X-ray pedal just before initial food-mouth contact (beginning of ingestion). The X-ray machines (Fig. 1b) were limited to 10 s of continuous exposure, after which an approximately 1 s break in recording was required before beginning a subsequent

trial. In most cases the complete feeding sequence (initial ingestion to terminal swallow) occurred within a single 10 s trial. Sometimes, however, one feeding sequence spanned multiple trials. In this study, 'trial' refers to the system-defined 7–10 s video, which often, but not always corresponded to a full feeding sequence. All trials contained a mix of gape cycle types[52] (i.e., stage 1 transport, rhythmic chewing, manipulation, stage 2 transport, swallow) that involved a range of tongue movements and shape changes that moved food through the mouth and into the esophagus. For select trials, as in Figs. 2 and 3, gape cycle types were determined via visual inspection of the X-ray videos in the open-source software XMALab (version 1.5) in accordance with commonly accepted definitions[52].

Multiple datasets (sessions comprising 40–60 trials of multiple food types), were collected for each subject across multiple days. However, due to inherent complexity and time-consuming nature of processing integrated XROMM and neural data, one session per subject was used in the present study. For each session 28 half-grape trials were drawn and partitioned according to the cross-validation scheme described below. Given the importance of across-session functionality in brain-machine interface-based protheses, future work should explore the stability of decoding across multiple days.

## XROMM data processing
We used the XROMM workflow to record and reconstruct the 3D rigid body motions of the cranium and mandible, as well as the 3D positions of a constellation of small beads implanted in the tongue (Fig. 1c; see refs. 16, 17 for a detailed description of the process). Kinematic data (biplanar X-ray videos to visualize radiopaque markers) were collected over multiple sessions at the University of Chicago XROMM Facility with Procapture 1.0.3.6 (Xcitex, Woburn, MA). Additionally, post-surgery CT scans were taken with a Vimago veterinary CT scanner (Epica Animal Health, Duncan, NC) from which mesh models of the cranium, mandible, and markers were created (segmented) in the open-source software 3D Slicer 4.11 (www.slicer.org). The 3D coordinates of the cranial and mandibular markers within each bone were extracted from the marker mesh models using the XROMM_MayaTools plug-in to enable rigid body fitting in XMALab.

Marker tracking was performed with a workflow[53] that integrates XMALab and DeepLabCut[54,55]. In short, deep neural networks were trained to track the 2D positions of the tantalum beads in both of the X-ray videos. Those 2D positions were then imported into XMALab where their 3D positions were triangulated, and the motion of the two rigid bodies (cranium and mandible) were computed. Rigid body transformation matrices and 3D points were filtered in XMALab with the built-in zero-lag, 30 Hz low-pass Butterworth filter. All subsequent data processing and analysis were performed in MATLAB 2020b (MathWorks, Natick, MA, U.S.A). All 3D imagery of tongue posture and jaw position were created and rendered with Maya 2020, except for Fig. 1d (made with MATLAB). In short, 3D bone models were generated from CT scans and were imported into Maya, where textures, lights, and virtual cameras were added. The final images were then exported either as still frames or as videos. Text was added to videos using Premiere Pro CC (Adobe Inc., San Jose, CA). The brain and monkey illustration seen in Fig. 1b, c, respectively, were generated with Illustrator CC (Adobe Inc., San Jose, CA).

## Kinematic variables
Jaw pitch was measured with a temporomandibular joint coordinate system[56], where the primary (i.e., first in rotation order) rotational axis passed through both mandibular condyles[57]. The joint coordinate system was computed by multiplying the mandible rigid body transformation matrix by the inverse of the cranium rigid body transformation matrix in every frame.

The first 7 tongue kinematic variables were calculated from the XYZ positions of subsets of the tongue markers. Sagittal flexion was the angle formed between the posterior deep, middle superficial, and tongue tip marker, following a recent definition[19]. Protrusion was the mean X-position value (relative to the cranium) of the three anterior-most markers (tongue tip, anterior superficial right and left). Roll was calculated using a pseudo-rigid body approach. First, the pseudo-rigid body motion of the anterior tongue, relative to the cranium, was calculated by fitting a rigid constellation of markers (taken from a frame at which the tongue was at rest), to the anterior 6 tongue markers in every frame of the video. Fitting was performed using MATLAB's Procrustes function, and the resultant rotation matrix was decomposed into Tait-Bryan angles, from which only x-axis rotation (roll) was extracted. The two lengths and widths were the Euclidean (straight-line) distances of pairs of markers (Fig. 1a inset; middle width, markers 5 and 6; anterior length, markers 1 and 4; middle length, markers 4 and 7; anterior width, markers 2 and 3).

Complex shape was quantified using a Generalized Procrustes Analysis approach[58]. In short, a constrained Procrustes superimposition (rigid transformation without reflection) was performed on the full constellation of tongue markers in every frame (all trials concatenated, for each individual), optimally fitting the tongue posture in each frame to a computed mean posture. This superimposition effectively removed changes to tongue position, rotation, and scale, leaving only changes in shape (deformation). Then a principal component analysis (PCA) was performed on the Procrustes-transformed *XYZ* marker positions (input: 7 markers, 21 dimensions), and the PC scores of the first 7 components (explaining 90%+ of the variance in tongue deformation) were used as "complex" deformation variables.

## Electrophysiology and neural data processing
Neural signals were recorded with Utah arrays (Blackrock Microsystems, Salt Lake City, UT) and Floating Microelectrode arrays (Microprobe for Life Science Inc, Gaithersburg, MD) using a Grapevine Neural Interface Processor (Ripple Neuro, Salt Lake City, UT). Signals were amplified and bandpass filtered between 0.1 Hz and 7.5 kHz, and recorded digitally (16-bit) at 30 kHz per channel. Only waveforms (1.7 ms in duration; 48 sample time points per waveform) that crossed a threshold were stored and offline spike sorted (Offline Sorter, Plexon, Dallas, TX) to remove noise and to isolate individual neurons. Total neuron counts were, for monkey Ry, 235 M1 neurons and 55 SC neurons, and for monkey Ye, 104 M1 neurons and 55 SC neurons. The time-varying firing rates of neurons were computed by summing spikes in 5 ms time bins (the same resolution as kinematic data). Preliminary analysis showed that decoding was possible with unsorted, multiunit activity but exhibited poorer performance than decoding with sorted neural data. Additional early analysis demonstrated that there was little difference in decoding performance between the two arrays in the same brain area (Supplementary Fig. 5), so data from the two M1 arrays and two SC arrays were combined in both subjects. For the analyses depicted in Figs. 2 and 3, 100 M1 neurons were randomly drawn from the pool of total neurons as decoder input for each subject.

## Decoding
We used a long short-term memory (LSTM) network to continuously decode tongue kinematics from cortical neuronal activity[21,22]. An LSTM network is a type of recurrent neural network where LSTM cells provide a means of mitigating the exploding/vanishing gradient problem through the selective "remembering" and 'forgetting' of specific information[59]. Here, we used MATLAB's native LSTM functionality in the Deep Learning Toolbox to train a series of LSTMs for sequence-to-sequence decoding. Input to the LSTM was a 2D array of binned spikes with dimensions number of neurons x number of timesteps. Output of the LSTM was the given predicted variable's values, in the form of an array with dimensions 1× number of timesteps. During inference (decoding of test trials), the network was provided with neural data in a stepwise manner, thus its instantaneous predictions were derived from

prior and present (but not future) neural activity. Network hyper-parameters are provided in Supplementary Table 2. We used a sevenfold cross-validation strategy to avoid overfitting. For each subject, 6-folds of the data were iteratively left out as test sets, and the seventh fold held out and used exclusively for hyperparameter selection. Each test fold comprised 4 trials of approximately 6–10 s in duration each. Each train fold comprised 24 trials of the same durations. For some analyses that required many iterations of training (single neuron decoding), three folds were randomly selected and used as test folds to minimize computation time (see Supplementary Table 2). To assess the likelihood of observing the high decoding performance of single neurons reported in Fig. 4c solely by chance, we performed an analysis in which we shuffled the neural data over feeding sequences, such that the neural data from feeding sequence X was used to decode the kinematics in sequence Y for 40 individual neurons with a mean firing rate of over 3 spikes/second. Shuffling was performed 10 times, and we then performed a permutation test (10,000 permutations) on the non-shuffled data and each iteration of shuffled data. To ensure the test yielded information about the right-tail of the distribution (see Fig. 4c), we used the mean of fourth quartile of the resampled data as the test statistic. In our results we conservatively report the maximum p-value of any shuffled iteration.

We performed Bayesian optimization-based hyperparameter selection for a subset of variables. In evaluating the optimization results, it immediately became evident that within a relatively large envelope the impact of changes to hyperparameters on decoding accuracy was minimal. This result is consistent with recent experimentation with network hyperparameter selection in decoding workflows[22]. In general, we found that increases in hidden unit number and epochs generally resulted in better accuracy, but those increases (e.g., an $R^2$ of 0.65 to 0.67 for hidden unit number increase of 200 to 400) were minimal relative to the heavy computational cost they incurred.

### Reporting summary

Further information on research design is available in the Nature Portfolio Reporting Summary linked to this article.

## Data availability

The raw neural and kinematic data are available on request from the corresponding author J.D.L.-C. Source data are provided with this paper.

## Code availability

The code used in the XROMM analysis of this study is available at https://doi.org/10.5281/zenodo.7734803. Additional MATLAB scripts on request from the corresponding author J.D.L.-C.

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

## Acknowledgements

We thank Rebecca Junod and Hernando Fereira for assistance in data collection and Eric Hosack, Victoria Hosack, Madison Jewell, Jared Luckas, Emma Lesser, Tricia Nicholson, and Derrick Tang for XROMM data processing assistance. We are deeply grateful to the veterinary staff of the University of Chicago Animal Resources Center for their constant care and support for the animals. Research reported in this publication was supported by National Institutes of Health grants from the National Institute of Dental and Craniofacial Research under Award Number R01DE027236 (F.I.A.-M., PI), by the National Institute On Aging under Award Number R01AG069227 (F.I.A.-M, PI), and by the National Institute of Neurological Disorders and Stroke under Award Number R01NS111982 (N.G.H. and C.F.R., co-PIs). The content is solely the responsibility of the authors and does not necessarily represent the official views of the National Institutes of Health. Additional funding was provided by the National Science Foundation Graduate Research Fellowship (to J.D.L.-C). Funding for the UChicago XROMM Facility was provided by National Science Foundation Major Research Instrumentation Grants MRI 1338036 and 1626552. This is University of Chicago XROMM Facility Publication #13.

## Author contributions

Conceptualization: F.I.A.-M., N.G.H., C.F.R.; methodology, all; investigation, all; software, J.D.L.-C; data analysis: J.D.L.-C.; data analysis supervision: F.I.A.-M., N.G.H; writing—original draft, J.D.L.-C; writing—review and editing, all; funding acquisition: F.I.A.-M. (PI), N.G.H., and C.F.R.; project administration and supervision, F.I.A.-M.

## Competing interests

The authors declare the following competing interests: N.G.H. serves as a consultant for BlackRock Microsystems, Inc., the company that sells the multi-electrode arrays implanted in sensorimotor cortices. The remaining authors declare no competing interests.
