## [Peer Review File · Nature Communications]

Robust cortical encoding of 3D tongue shape during feeding in macaquesREVIEWER COMMENTS

Reviewer #1 (Remarks to the Author):

In this manuscript, Laurence-Chasen and colleagues used simultaneous single unit recordings of sensorimotor cortex and intraoral X-ray videography to track tongue kinematics during eating. From this data, they were able to accurately decode tongue kinematics and shape from motor cortex neurons using machine learning. The use of X-ray videography to track the complex and otherwise inaccessible movements of the tongue is innovative and has the potential to pave the way for brain computer interfaces to restore tongue function. This paper will be of general interest once the authors address the following concerns:

Major

1. The authors should provide sufficient details of their experimental techniques to allow others to reproduce the data.
 - a. Please provide specific locations of where the marker beads were implanted and the rationale for choosing the specific 7 of 15 beads. Similarly, please report which marker pairs were used for calculating Euclidean distances to extract the regional lengths and widths.
 - b. Please provide the coordinates used for placing the electrodes as well as histologic/MRI confirmation of their location.
2. More characterization of the behavioral data is needed to interpret the decoding. How frequently did the 7 kinematic coded variables, 3 kinematic PCs, and 7 shape PCs occur over the course of the recording session? The authors should also add the first 3 PCs of kinematic variables to figure 2.
3. Along similar lines, weighting each of the 14 components equally to compute the average decoding may not be valid (figure 4b), as each component does not contribute equally to the tongue movements during eating. The kinematic and shape variable decoding should also be averaged separately and added to figures 2 and 3, respectively.
4. To demonstrate the effectiveness of the decoding, the authors should show the RMSE for tongue kinematics (as they do for shape in extended figure 3).
5. The authors should characterize whether the individual single units in motor and sensory cortex are significantly modulated by tongue movements/shape and whether such significantly modulated neurons

contain disproportionate decoding information. This analysis might provide insight into the differences between decoding accuracies across individuals and cortex type (sensory vs motor).

6. Figure 4 shows high-performance single units that accurately decode tongue deformation. To evaluate this result, it would be helpful to have a statistical measure that demonstrates the likelihood of such neurons arising by chance.

Minor

Figure 1e does not add any information of value and should be removed.

Referenced supplemental video seemed to not be provided.

Reviewer #2 (Remarks to the Author):

In this study, the authors combine x-ray tracking of tongue and population decoding to examine the representation of 3D tongue shape in motor cortex of monkeys. X-ray imaging of multiple beads implanted in the tongue tracks the 3D position. Tongue kinematics and shape are derived analytically from movement and deformation. Neural activity from orofacial representations of M1 and S1 is recorded with Utah arrays. Decoding analysis shows that both 3D tongue kinematics and shape are represented in neural activity, and more so in M1 than S1. This study presents interesting data on an understudied and important area of motor control. The manuscript is well written, and the finding that motor cortex represents 3D tongue deformation will be of interest to many. We only have clarification questions which can be address through a few additional analysis and changes to the text.

Major

1) A key conclusion of the paper is that M1 represents 3D tongue shape information. The results show robust decoding of tongue shape changes, yet we have a few questions about the decoding results.

a. Tongue shape is correlated with tongue kinematics (Extended Data Fig 1). The correlation is higher for PCs with better decoding. It would be important to decorrelate tongue shape from kinematics, either explicitly or to control for this using some analysis similar to Extended Data Fig 4 to show that the decoding is not driven by information about tongue kinematics.

b. It is somewhat difficult to judge what the tongue shape changes correspond to. Can authors discuss whether the tongue shape changes are directed toward specific behaviors (e.g. swallowing, chewing), and whether the information may differ between different behaviors. For example, do authors observe differences in decoding for different behavior epochs?

2) The comparison between M1 and S1 is informative and important. We have two clarification questions.

a. Where are the array locations within S1 and M1? How do the authors know if the arrays target orofacial representations? As the authors discussed, the difference in decoding accuracy between the two monkeys could be due to difference in array placement. Extended Data Fig 5 also shows decoding differences. But it is not clear if these are related to array location. In general, we could not find any information regarding the location of the recordings, either based on functional mapping or anatomical location. Having this information would be helpful.

b. Can the authors indicate which variable is which in Fig 4a (kinematics vs. shape)? Breaking down Fig 4b for individual variables will be informative, perhaps as a supplemental figure.

3) It is somewhat difficult to judge how much data is present in this study from each monkey. Judging from the Method description about training and testing, the data seems to be 28 trials for each subject (24 trial for training and 4 trials for testing). Are these trials collected across different sessions? Ideally, it could be shown that decoding within each subject is replicated across sessions, since the application to brain machine interface-base prostheses for restoring orolingual function and communication is an implication and is mentioned in the Discussion. The authors should provide information about decoding accuracy across different sessions of recordings. If the setup of electrophysiological recording with XROMM is particularly difficult and the data is limited, at the very least it should be clarified how many sessions are in each monkey and this limitation should be mentioned.

4) Some aspects of the Methods are missing:

a. Extend Data Fig 4, how is Jaw position derived from the position of the beads that are implemented into cranium & mandible?

b. Figs 2a and 3a, how are the different behavior epochs defined?

c. More information about the behavior is needed. For example, what is “stage 2 transport”? what is “manipulation”? What epochs constitute a “trial”?

Minor

1) The main justification for the use of LSTM is its accuracy and being a nonlinear decoder. Can the authors provide some comparisons to a linear decoder (e.g. GLM)?

2) Does the cleanness of spike sorting affect the decoding accuracy? For example, can unsorted multi-unit activity decode just as well?

3) Fig 4b, what is the error bar? It is not described in the legend.

Reviewer #3 (Remarks to the Author):

The manuscript of Laurence-Chasen and colleagues uses a combination of experimental and computational techniques to relate the cortical activity in the somatosensory cortex and primary motor cortex of primates to movement and shape changes of the tongue. Tongue deformation has been difficult to quantify due to the inability to track intraoral changes. A description of how this 3-d soft-body is embedded and controlled by cortical activity could advance the implementation of orofacial neuromechanics and soft robotics.

The authors used x-ray video technology to obtain a 3-d model of bone morphology (in this case the cranium and mandible) and to track the motion of 7 radioopaque markers in the tongue. Together, this technique allowed for the 3-d reconstruction of the 7 tongue markers relative to the cranium. Cortical activity has been linked to 2-d movement of such markers, but this may be the first study that decodes 3-d tongue kinematics and 3-d tongue-shape. The authors concurrently record activity from large neural populations in sensorymotor and motor cortex, and then use multi-dimensional decoders to relate neural activity to tongue kinematics and shape.

Beyond the experimental contributions, which may well be novel and state-of-the-art, the computational results mostly seem to amount to a proof of concept that 3-d tongue movements can be decoded from the relevant cortical areas. These computational results, however, at present lack a clear framing (what other outcome would have been possible?) and motivation (e.g. why the particular use of decoders?) and several aspects of the employed methodologies and results are not adequately discussed. Overall, I was left with the impression of a potentially interesting work that remains somewhat superficial, and whose relevance and soundness are difficult to evaluate.

Main comments

(1) There is some tension in the paper about the goal of the work: is the goal to obtain a decoder that works as well as possible, as a step towards a prosthetic tongue? Or is the goal to uncover general principles of brain function and motor control? Both are mentioned repeatedly, but after reading the paper it was neither clear to me if the findings (1) could be used, in the form they are presented here, for practical applications such as developing orofacial neuromechanics or brain-machine interface-based prostheses, or (2) how they advance our understanding of motor principles.

(2) specifically: what are the common motor control principles that the authors mention throughout the paper? How are these principles related to the finding that several tongue parameters are decodable from the neural population? is it surprising that the considered parameters can be decoded, considering that the targeted motor areas were known to be important for tongue movements? The authors should explicitly state what alternative outcomes would have been possible that would have pointed to differences in the motor principles governing movements of limbs and tongues.

(3) There is no clear justification for why a non-linear decoder was favored over a linear one. The structure of a linear decoder would potentially be easier to understand. In a previous paper ('Decoding hand kinematics from population responses in sensorimotor cortex during grasping' from the same lab), the authors tested a series of decoders and found that the non-linear ones did not substantially improve the decoding. In this previous paper, the favored decoder was a Kalman filter fitted to recorded joints. The Kalman filter comprised two estimates of some variable, one based on past kinematics and the other based on the current neural activity. For hand kinematics, the variable was joint angle or joint angular velocity. In the current case, the variable could be any of the tongue-related variables. If the authors tried simpler decoders and found them unsuitable for these data, I think it is necessary to mention what was done and what were the challenges.

(4) What are the inputs and outputs to the LSTM? Based on the Matlab description of the function, I would expect it to be the spike raster with dimensionality No. of neurons x Time-window. This should be explicitly mentioned.

(5) The choice of decoding of PC coefficients as opposed to measured variables of tongue-shape should be better justified. One novel finding of this paper is arguably that tongue-shape may be represented in cortical activity. However, I see the following confounds. First, it is only the PC coefficients that are decoded. It is not clear why the neural activity would be modulated by this compressed arbitrary transformation of the tongue shape. Second, out of the 7 considered PCs, all except PC7, have correlations above 0.36 with kinematic variables (PC1 & sagittal flexion-0.77; PC2 & middle width - 0.42; PC3 & middle width - 0.36; PC4 & anterior length - 0.66; PC5 & middle width - 0.43; PC6 & middle width - 0.38). These correlations blur the significance of the result that quantifies to what extent the neural activity represents tongue-shape, as opposed to kinematic variables. Third, Extended Figure 3 illustrates that PC5-7 do not further decrease the reconstruction error, an effect which: (1) seems to indicate that these last three PCs do not capture a portion of the neural variance that is meaningful to tongue-shape; and (2) is not explicitly mentioned in the main text (line 118). Finally, a comment on how surprising (or unsurprising) is the fact that the 7PCs in Extended Figure 2 are not consistent across the two monkeys is missing.

Possible questions related to neural representations that could be addressed:

(6) To better describe tongue-related representations, the relation between kinematic-related subspace and tongue-shape subspace could be considered. One proposal is to extend the analysis of Figure 4 to describe whether the different decoded variables reveal mixed or structured representations at the level of single neurons. Specifically, to look how each neuron represents (in the sense that it carries information sufficient to decode) all the considered variables. Are there neurons that encode all variables or most neurons would only represent just a few, but not all, variables?

(7) The relation between the considered kinematic and tongue-shape variables could be better quantified. First, how well can one decode the behavior-type (swallow, chew, manipulate) from the behavior, i.e. a. the kinematics; b. the tongue-shape. This would add some functional significance to the various tongue-related variables. Second, is there a certain behavior-type (swallow, chew, manipulate) where the kinematic and tongue-shape variables are better decoded? For example, are reflexive components of the behavior decoded less well?

(8) The authors comment that tongue roll is the best decoded kinematic variable. Since tongue kinematics are differently correlated with the jaw (Extended Fig. 1), a potentially fairer comparison of which kinematic is best decoded would be to evaluate only segments (Extended Fig. 4) where this correlation is small.

Minor comments on the Supplementary Material

(9) line 103. Fig. 1B displays the electrode array and does not contain any length and width.

(10) line 107. Fig. 2C does not exist. Is the mean posture displayed somewhere else?

(11) Explicit description of input data, i.e. dimensionality of kinematics and shape data, for PCA analysis.

(12) Is the behavioral data (XYZ markers) processed in any way, such as for example smoothed?

Minor comments on the Main text

Line 74. Reference should be added to 'standard tongue kinematic variables'.

We thank all of the reviewers for their feedback and have responded to their specific comments in blue. Of note, we have added several new supplementary figures that support new analyses and have re-organized the manuscript to meet the journal's formatting guidelines.

Reviewer #1 (Remarks to the Author):

In this manuscript, Laurence-Chasen and colleagues used simultaneous single unit recordings of sensorimotor cortex and intraoral X-ray videography to track tongue kinematics during eating. From this data, they were able to accurately decode tongue kinematics and shape from motor cortex neurons using machine learning. The use of X-ray videography to track the complex and otherwise inaccessible movements of the tongue is innovative and has the potential to pave the way for brain computer interfaces to restore tongue function. This paper will be of general interest once the authors address the following concerns:

We thank the reviewer for their thoughtful comments.

Major

1. The authors should provide sufficient details of their experimental techniques to allow others to reproduce the data.
 - a. Please provide specific locations of where the marker beads were implanted and the rationale for choosing the specific 7 of 15 beads. Similarly, please report which marker pairs were used for calculating Euclidean distances to extract the regional lengths and widths.

Details of marker locations, rationale for the chosen subset (Figure 1a, Supplementary Methods lines 16-21), and pairs used for calculated distances have been added to the methods. (Supplementary Methods lines 98-100)

- b. Please provide the coordinates used for placing the electrodes as well as histologic/MRI confirmation of their location.

We have added a line drawing illustrating the location of the arrays relative to sulcal landmarks for each monkey. (Supplementary Fig. 8) We also added notes to the Supplementary methods detailing 1) our multi-modal approach to array surgical planning, 2) that during surgical implantation we used surface electrical stimulation to identify orofacial motor cortex, and 3) a method of post-op confirmation of location. (Supplementary Methods Lines 24-42).

2. More characterization of the behavioral data is needed to interpret the decoding. How frequently did the 7 kinematic coded variables, 3 kinematic PCs, and 7 shape PCs occur over the course of the recording session? The authors should also add the first 3 PCs of kinematic variables to figure 2.

As feeding involves cyclic motions of the jaw and tongue (punctuated by discrete swallows), all of the decoded variables 'occurred' frequently and continuously over the duration of the feeding sequences. We separate kinematic and shape variables in figures 2 and 3, respectively, for conceptual distinction

and to avoid over-crowding of the figures. As the figures share the same time-scale and source trial, readers, if they desire, will be able to compare the two.

3. Along similar lines, weighting each of the 14 components equally to compute the average decoding may not be valid (figure 4b), as each component does not contribute equally to the tongue movements during eating. The kinematic and shape variable decoding should also be averaged separately and added to figures 2 and 3, respectively.

We agree with the reviewer that average decoding performance across different components is not statistically sound--for this reason we have separated the two variable types in figure 4B and weighted the shape variables (PCs) according to their fraction of variance explained when averaging across variables.

Additionally, we do not believe that figures 2 and 3 require an average across variables, as the current format allows the reader to assess the decoding performance at the level they desire (individual variables, groups of variables).

4. To demonstrate the effectiveness of the decoding, the authors should show the RMSE for tongue kinematics (as they do for shape in Supplementary figure 3).

A supplementary figure showing RMSE for tongue kinematics has been added (Supplementary Fig. 6)

5. The authors should characterize whether the individual single units in motor and sensory cortex are significantly modulated by tongue movements/shape and whether such significantly modulated neurons contain disproportionate decoding information. This analysis might provide insight into the differences between decoding accuracies across individuals and cortex type (sensory vs motor).

We agree that an in-depth analysis at the single-unit level is a clear next step, but we believe that with the extensive additional analyses added in this revision this is beyond the scope of this manuscript.

6. Figure 4 shows high-performance single units that accurately decode tongue deformation. To evaluate this result, it would be helpful to have a statistical measure that demonstrates the likelihood of such neurons arising by chance.

Following this suggestion, we performed an additional analysis in which we shuffled the neural data over feeding sequences, such that the neural data from feeding sequence X was used to decode the kinematics in sequence Y for 40 individual neurons with a mean firing rate of over 3 spikes/second.

In our non-shuffled data we found 38 and 26 variables that were decoded from single neurons with an R^2 above 0.15 for Monkey Ry and Ye, respectively. We shuffled the data 10 times and found that the mean number of variables decoded with an R^2 in a similar range was 8.33 ± 1.7 s.e. and 0, respectively. Raising the threshold to R^2 over .25, the non-shuffled counts were 7 and 6, and the mean shuffled counts

were 2.4 ± 0.66 s.e. and 0, respectively. Thus we conclude that the likelihood of this number of neurons and variables emerging by chance is exceedingly low.

Minor

Figure 1e does not add any information of value and should be removed.

We respectfully disagree and believe it is worthwhile to include the spike raster because it is only place in the paper where we show raw neural data, showing how orofacial motor cortical neurons fire during feeding.

Referenced Supplementary video seemed to not be provided.

We have added the missing Supplementary video.

Reviewer #2 (Remarks to the Author):

In this study, the authors combine x-ray tracking of tongue and population decoding to examine the representation of 3D tongue shape in motor cortex of monkeys. X-ray imaging of multiple beads implanted in the tongue tracks the 3D position. Tongue kinematics and shape are derived analytically from movement and deformation. Neural activity from orofacial representations of M1 and S1 is recorded with Utah arrays. Decoding analysis shows that both 3D tongue kinematics and shape are represented in neural activity, and more so in M1 than S1. This study presents interesting data on an understudied and important area of motor control. The manuscript is well written, and the finding that motor cortex represents 3D tongue deformation will be of interest to many. We only have clarification questions which can be address through a few additional analysis and changes to the text.

We thank the reviewer for their thoughtful feedback.

Major

1) A key conclusion of the paper is that M1 represents 3D tongue shape information. The results show robust decoding of tongue shape changes, yet we have a few questions about the decoding results.

a. Tongue shape is correlated with tongue kinematics (Supplementary Fig 1). The correlation is higher for PCs with better decoding. It would be important to decorrelate tongue shape from kinematics, either explicitly or to control for this using some analysis similar to Supplementary Fig 4 to show that the decoding is not driven by information about tongue kinematics.

This is an important point, and to this end, we have performed a new analysis similar to Supplementary Fig 4, where we quantified the shape decoding performance during 1-s regions of low correlation with three of the kinematic variables (sagittal flexion, protrusion, and roll), see Supplementary Fig. 9. As the figure illustrates, there are some shape variables whose relatively high decoding performance can be

attributed to correlations with the kinematic variables (Shape PC1 for both monkeys). However, both subjects have shape variables that were decoded well, even when weakly correlated with multiple kinematic variables (see Monkey Ry, Shape PC 4; Monkey Ye, Shape PC2). The persistence of relatively high decoding accuracy, even in periods where the tongue's shape change is weakly correlated with its overall movements suggests a cortical representation of shape. That some of the shape decoding is attributable to correlations with the kinematic variables isn't surprising, given lingual anatomy; many of the same muscles are both moving and deforming the tongue, and thus fully de-correlating kinematics and shape isn't necessarily tractable.

b. It is somewhat difficult to judge what the tongue shape changes correspond to. Can authors discuss whether the tongue shape changes are directed toward specific behaviors (e.g. swallowing, chewing), and whether the information may differ between different behaviors. For example, do authors observe differences in decoding for different behavior epochs?

We agree that the question of differences in decoding across different behaviors (chew vs. swallowing) is very interesting and a clear next step. We feel that this is beyond the scope of this manuscript at present.

2) The comparison between M1 and S1 is informative and important. We have two clarification questions.

a. Where are the array locations within S1 and M1? How do the authors know if the arrays target orofacial representations? As the authors discussed, the difference in decoding accuracy between the two monkeys could be due to difference in array placement. Supplementary Fig 5 also shows decoding differences. But it is not clear if these are related to array location. In general, we could not find any information regarding the location of the recordings, either based on functional mapping or anatomical location. Having this information would be helpful.

We have added an additional figure showing the exact array locations by individual (Supplementary Fig. 8) and have included the clarification in the text that surface stimulation was used to locate the tongue region of the orofacial sensorimotor cortex (Supplementary Methods lines 30-31).

b. Can the authors indicate which variable is which in Fig 4a (kinematics vs. shape)? Breaking down Fig 4b for individual variables will be informative, perhaps as a Supplementary figure.

We have altered Fig 4a to indicate which variable is kinematics vs. shape, and have also broken down Fig 4b into kinematics and shape variables, for clarity and based on Reviewer 3's concern about a lack of weighting of principal components.

3) It is somewhat difficult to judge how much data is present in this study from each monkey. Judging from the Method description about training and testing, the data seems to be 28 trials for each subject (24 trial for training and 4 trials for testing). Are these trials collected across different sessions? Ideally, it could be shown that decoding within each subject is replicated across sessions, since the application to brain machine interface-base prostheses for restoring orolingual function and communication is an implication and is mentioned in the Discussion. The authors should provide information about decoding accuracy across different sessions of recordings. If the setup of electrophysiological recording with XROMM is particularly difficult and the data is limited, at the very least it should be clarified how many sessions are in each monkey and this limitation should be mentioned.

Both XROMM/e-phys setup and data processing is exceptionally challenging, and thus data from one recording session were used in this study. We have revised the “Behavioral Task” section of the Supplementary methods to be “Behavioral Task and Dataset Composition” and have provided details the exact amount of data used in this study and noted this limitation. (Supplementary Methods lines 44-64)

4) Some aspects of the Methods are missing:

a. Extend Data Fig 4, how is Jaw position derived from the position of the beads that are implemented into cranium & mandible?

We have added a reference in the figure caption to the methods section that describes the process by which mandibular kinematics are reconstructed from 3D bead positions (Supplementary methods lines 74-91).

b. Figs 2a and 3a, how are the different behavior epochs defined?

We added references in the figure legends to new text in the Supplementary methods that describes how we defined gape cycle types via visual inspection of the x-ray videos following commonly accepted definitions (Supplementary methods lines 55-58).

c. More information about the behavior is needed. For example, what is “stage 2 transport”? what is “manipulation”? What epochs constitute a “trial”?

We added more explicit language that links the specific gape cycle types to the processes they involve (Main Text Lines 63-65), and clarified that trials involve a mix of the different gape cycle types (Supplementary methods lines 53-55)

Minor

1) The main justification for the use of LSTM is its accuracy and being a nonlinear decoder. Can the authors provide some comparisons to a linear decoder (e.g. GLM)?

We added the results from a standard linear decoder--the Kalman filter (Supplementary Fig. 7). Its performance is substantially worse than the LSTM.

2) Does the cleanness of spike sorting affect the decoding accuracy? For example, can unsorted multi-unit activity decode just as well?

This is a great question. We found that we could successfully decode with unsorted activity, but the performance was not as good. We added a note of this to the text (Supplementary methods lines 123-124)

3) Fig 4b, what is the error bar? It is not described in the legend.

We clarified that the error region represents ± 1 standard deviation in the legend.

Reviewer #3 (Remarks to the Author):

The manuscript of Laurence-Chasen and colleagues uses a combination of experimental and computational techniques to relate the cortical activity in the somatosensory cortex and primary motor cortex of primates to movement and shape changes of the tongue. Tongue deformation has been difficult to quantify due to the inability to track intraoral changes. A description of how this 3-d soft-body is embedded and controlled by cortical activity could advance the implementation of orofacial neuromechanics and soft robotics.

The authors used x-ray video technology to obtain a 3-d model of bone morphology (in this case the cranium and mandible) and to track the motion of 7 radioopaque markers in the tongue. Together, this technique allowed for the 3-d reconstruction of the 7 tongue markers relative to the cranium. Cortical activity has been linked to 2-d movement of such markers, but this may be the first study that decodes 3-d tongue kinematics and 3-d tongue-shape. The authors concurrently record activity from large neural populations in sensorymotor and motor cortex, and then use multi-dimensional decoders to relate neural activity to tongue kinematics and shape.

Beyond the experimental contributions, which may well be novel and state-of-the-art, the computational results mostly seem to amount to a proof of concept that 3-d tongue movements can be decoded from the relevant cortical areas. These computational results, however, at present lack a clear framing (what other outcome would have been possible?) and motivation (e.g. why the particular use of decoders?) and several aspects of the employed methodologies and results are not adequately discussed. Overall, I was left with the impression of a potentially interesting work that remains somewhat superficial, and whose relevance and soundness are difficult to evaluate.

We thank the reviewer for their thoughtful comments and conceptual challenges.

Main comments

(1) There is some tension in the paper about the goal of the work: is the goal to obtain a decoder that works as well as possible, as a step towards a prosthetic tongue? Or is the goal to uncover general principles of brain function and motor control? Both are mentioned repeatedly, but after reading the paper it was neither clear to me if the findings (1) could be used, in the form they are presented here, for practical applications such as developing orofacial neuromechanics or brain-machine interface-based prostheses, or (2) how they advance our understanding of motor principles.

We believe the primary goal of this study is to advance our understanding of motor principles. In particular, feeding is semi-automatic and is typically assumed to involve structures outside of cortex. The fact that M1 and SC can decode details of tongue shape suggest at a minimum that sensorimotor cortex is informed of the detailed kinematics and shape of the tongue during feeding. It may indicate that the cortex is involved in driving behavior in a soft-tissue effector. This work also provides foundational information that detailed tongue kinematics and shape can be predicted from cortex which has practical implications for feeding and language neural prostheses.

(2) specifically: what are the common motor control principles that the authors mention throughout the paper? How are these principles related to the finding that several tongue parameters are decodable from the neural population? is it surprising that the considered parameters can be decoded, considering that the targeted motor areas were known to be important for tongue movements? The authors should explicitly state what alternative outcomes would have been possible that would have pointed to differences in the motor principles governing movements of limbs and tongues.

The tongue is not a rigid structure with joints and can take on a multitude of configurations. The fact that decoding is comparable for tongue and limbs is not a priori expected. An alternate outcome would have been that sensorimotor cortex decodes gross movements of the tongue such as flexion and protrusion/retraction, but not detailed tongue shape.

(3) There is no clear justification for why a non-linear decoder was favored over a linear one. The structure of a linear decoder would potentially be easier to understand. In a previous paper ('Decoding hand kinematics from population responses in sensorimotor cortex during grasping' from the same lab), the authors tested a series of decoders and found that the non-linear ones did not substantially improve the decoding. In this previous paper, the favored decoder was a Kalman filter fitted to recorded joints. The Kalman filter comprised two estimates of some variable, one based on past kinematics and the other based on the current neural activity. For hand kinematics, the variable was joint angle or joint angular velocity. In the current case, the variable could be any of the tongue-related variables. If the authors tried simpler decoders and found them unsuitable for these data, I think it is necessary to mention what was done and what were the challenges.

We added the results from a standard linear decoder--the Kalman filter (Supplementary Fig. 7). Its

performance is substantially worse than the LSTM.

(4) What are the inputs and outputs to the LSTM? Based on the Matlab description of the function, I would expect it to be the spike raster with dimensionality No. of neurons x Time-window. This should be explicitly mentioned.

Thank you; your expectation is exactly right, and we've added that text (Supplementary Methods lines 135-137)

(5) The choice of decoding of PC coefficients as opposed to measured variables of tongue-shape should be better justified. One novel finding of this paper is arguably that tongue-shape may be represented in cortical activity. However, I see the following confounds. First, it is only the PC coefficients that are decoded. It is not clear why the neural activity would be modulated by this compressed arbitrary transformation of the tongue shape.

We respectfully disagree with the characterization that PC coefficients represent a totally “arbitrary” transformation of the tongue shape. While PCs do not *necessarily* relate to biological movement in a strict sense, we found the first several PC components correspond biologically intuitive aspects of tongue deformation (Supplementary Fig 2). Moreover, there is clear precedent in the literature in decoding PCs as a means of exploring muscle synergies and potential cortical control systems (Okorokova et al., 2020; Mollazadeh et al., 2014). As a soft body with many muscles, it is possible that the brain relies on a compressed representation of tongue deformation to rapidly deform and move it in the mouth.

Second, out of the 7 considered PCs, all except PC7, have correlations above 0.36 with kinematic variables (PC1 & sagittal flexion-0.77; PC2 & middle width - 0.42; PC3 & middle width - 0.36; PC4 & anterior length - 0.66; PC5 & middle width - 0.43; PC6 & middle width - 0.38). These correlations blur the significance of the result that quantifies to what extent the neural activity represents tongue-shape, as opposed to kinematic variables.

This is a fundamental challenge in this system; it is not clear how one could completely decorrelate tongue shape from tongue kinematics. To better address this problem, we present a new analysis in Supplementary Fig. 9. where we quantified the shape decoding performance during 1-s regions of low correlation with three of the kinematic variables (sagittal flexion, protrusion, and roll). Here is the same interpretation we provided in response to Reviewer 2's similar comments:

As the figure illustrates, there are some shape variables whose relatively high decoding performance can be attributed to correlations with the kinematic variables (Shape PC1 for both monkeys). However, both subjects have shape variables that were decoded well, even when weakly correlated with multiple kinematic variables (see Monkey Ry; Shape PC 4, Monkey Ye, Shape PC2.) The persistence of relatively high decoding accuracy, even in periods where the tongue's shape change is weakly correlated with its overall movements suggests a cortical representation of shape. That some of the shape decoding is attributable to correlations with the kinematic variables isn't surprising, given lingual anatomy; many of

the same muscles are both moving and deforming the tongue, and thus fully de-correlating kinematics and shape isn't necessarily tractable.

Third, Supplementary Figure 3 illustrates that PC5-7 do not further decrease the reconstruction error, an effect which: (1) seems to indicate that these last three PCs do not capture a portion of the neural variance that is meaningful to tongue-shape; and (2) is not explicitly mentioned in the main text (line 118). Finally, a comment on how surprising (or unsurprising) is the fact that the 7PCs in Supplementary Figure 2 are not consistent across the two monkeys is missing.

We have added language to the text about the result that the last PCs may not capture a portion of neural variance that is meaningful to tongue shape (Main Text lines 122-126) as well as commentary about the consistency of results across the two monkeys (Main Text 205-210)

Possible questions related to neural representations that could be addressed:

(6) To better describe tongue-related representations, the relation between kinematic-related subspace and tongue-shape subspace could be considered. One proposal is to extend the analysis of Figure 4 to describe whether the different decoded variables reveal mixed or structured representations at the level of single neurons. Specifically, to look how each neuron represents (in the sense that it carries information sufficient to decode) all the considered variables. Are there neurons that encode all variables or most neurons would only represent just a few, but not all, variables?

This is a good suggestion, and to this end we have generated a Supplementary figure that depicts the decoding results for three high-performing neurons per subject, by variable (Supplementary Fig. 10). The figure illustrates that some neurons perform well on multiple variables (Monkey Ry, Neuron A) whereas others perform well on only one (Monkey Ry Neuron B).

(7) The relation between the considered kinematic and tongue-shape variables could be better quantified. First, how well can one decode the behavior-type (swallow, chew, manipulate) from the behavior, i.e. a. the kinematics; b. the tongue-shape. This would add some functional significance to the various tongue-related variables. Second, is there a certain behavior-type (swallow, chew, manipulate) where the kinematic and tongue-shape variables are better decoded? For example, are reflexive components of the behavior decoded less well?

We agree that decoding behavior type from tongue shape and linking decoding performance to different behaviors are both very interesting and clear next steps; we have added text to the discussion explicitly mentioning this (Main text lines 210-212). However, we feel that this is beyond the scope of this manuscript at present.

(8) The authors comment that tongue roll is the best decoded kinematic variable. Since tongue kinematics are differently correlated with the jaw (Supplementary Fig. 1), a potentially fairer comparison

of which kinematic is best decoded would be to evaluate only segments (Supplementary Fig. 4) where this correlation is small.

See earlier response and Supplementary Fig. 9.

Minor comments on the Supplementary Material

(9) line 103. Fig. 1B displays the electrode array and does not contain any length and width.

We corrected this section to reflect changes to the figure that specify which markers were used for each measurement (Supplementary methods lines 102-104).

(10) line 107. Fig. 2C does not exist. Is the mean posture displayed somewhere else?

We removed the reference to figure 2C. The mean posture is best represented by the pose depicted in Fig. 1A.

(11) Explicit description of input data, i.e. dimensionality of kinematics and shape data, for PCA analysis.

We added the dimensionality (Supplementary methods lines 111)

(12) Is the behavioral data (XYZ markers) processed in any way, such as for example smoothed?

We describe filtering in Supplementary Methods: XROMM Data processing (lines 82-83) "Rigid body transformation matrices and 3D points were filtered in XMALab with the built-in zero-lag, 30 Hz low-pass Butterworth filter"

Minor comments on the Main text

Line 74. Reference should be added to 'standard tongue kinematic variables'.

We added a reference to Feilich et al. 2021 (line 76)

REVIEWERS' COMMENTS

Reviewer #1 (Remarks to the Author):

The authors' response generally addresses my concerns. Well done!

However, for point 6, shuffling the data is fine, but they should add an actual statistical evaluation to the text (i.e., with a corresponding p value) indicating what is the likelihood of finding the high-performing single M1o neurons shown in 4c by chance. The analysis the authors presented in the author response does not do that.

Secondly, while I am all for showing raw neural data, the raster plot in figure 1e doesn't show "how orofacial motor cortical neurons fire during feeding." It shows a raster plot of 100 neurons firing over 1.5 seconds. What does time zero represent? There is no indication from the raster plot that the neurons are firing in any behaviorally linked fashion. It could just as easily be any neurons firing during any epoch. Perhaps the authors can indicate behaviorally pertinent epochs on the raster to highlight the neurons responding to key kinematic events.

Reviewer #2 (Remarks to the Author):

The authors have addressed my comments. The added decoding analysis controlling for correlations between tongue kinematics and shape variables is a great addition. The revised methods is much improved. The finding that motor cortex represents 3D tongue deformation will be a valuable addition to an understudied area of motor control research. I support publication in Nature Communications.

Reviewer #3 (Remarks to the Author):

The authors made several changes to the text and added analyses that address most of my concerns about the original submission.

I do think that the authors could still be somewhat more explicit about the implications of their results. The comparison of decoding performance between kinematic variables and PCs is interesting, and I appreciate the authors' point that interpreting the results is challenging because of the correlations between variables and PCs. Nonetheless, the results create a bit of a "glass-half-empty or glass-half-full" problem. One outcome could have been that decoding is overall much better for PCs than for kinematic variables, or vice-versa. That is not what the authors find. Decoding performance is similar between the two, and if anything, somewhat higher for the best kinematic variables. The new Supp. Fig. 9 suggests that some aspects of the neural responses are not captured by the kinematic variables (although that figure would be more informative if it also showed what "weakly-correlated" for the kinematic variables actually means in terms of a number). But an alternative reading of that figure would be that most PCs are substantially less well decodable in the periods of "weak-correlation", suggesting a weak encoding of tongue shape. So overall I am not sure that the data strongly supports the current (implied) statement that "sensorimotor cortex [encodes ...] detailed tongue shape" (line 184).

Considering the difficulty inherent in interpreting the behavioral relevance of the PCs, and in comparing them across monkeys, I would recommend to show Supp. Fig. 1 separately for the two monkeys. Any differences across monkeys in the exact ordering/functional relevance of individual PCs could lead to overall smaller "peak" correlation values between kinematic variables and PCs when averaging across monkeys.

We thank all of the reviewers for their continued feedback and have responded to their comments in and noted changes in the main text in blue. We have supplied raw data for the figures and ensured that all individual data points are plotted in the case of bar plots.

We also substantially revised Supplemental Fig. 9, which we now believe much more clearly illustrates decoding of kinematics-independent tongue shape.

And one last minor note, in a code review, we found 1 of the 6 cross-validation folds was being unintentionally omitted during mean calculations for some figures, thus there are minor changes to values in Figs. 2b, 3b, and 4a that do not impact overall trends or interpretation.

REVIEWERS' COMMENTS

Reviewer #1 (Remarks to the Author):

The authors' response generally addresses my concerns. Well done!

However, for point 6, shuffling the data is fine, but they should add an actual statistical evaluation to the text (i.e., with a corresponding p value) indicating what is the likelihood of finding the high-performing single M1o neurons shown in 4c by chance. The analysis the authors presented in the author response does not do that.

We performed a modified permutation test that provided a p value associated with these results ($p < 0.0001$) and added it to the text. Lines: 154-157 and 372-381.

Secondly, while I am all for showing raw neural data, the raster plot in figure 1e doesn't show "how orofacial motor cortical neurons fire during feeding." It shows a raster plot of 100 neurons firing over 1.5 seconds. What does time zero represent? There is no indication from the raster plot that the neurons are firing in any behaviorally linked fashion. It could just as easily be any neurons firing during any epoch. Perhaps the authors can indicate behaviorally pertinent epochs on the raster to highlight the neurons responding to key kinematic events.

We added behavioral epochs to the raster.

Reviewer #2 (Remarks to the Author):

The authors have addressed my comments. The added decoding analysis controlling for correlations between tongue kinematics and shape variables is a great addition. The revised methods is much improved. The finding that motor cortex represents 3D tongue deformation will be a valuable addition to an understudied area of motor control research. I support publication in Nature Communications.

Thank you very much!

Reviewer #3 (Remarks to the Author):

The authors made several changes to the text and added analyses that address most of my concerns about the original submission.

I do think that the authors could still be somewhat more explicit about the implications of their results. The comparison of decoding performance between kinematic variables and PCs is interesting, and I appreciate the authors' point that interpreting the results is challenging because of the correlations between variables and PCs. Nonetheless, the results create a bit of a "glass-half-empty or glass-half-full" problem. One outcome could have been that decoding is overall much better for PCs than for kinematic variables, or vice-versa. That is not what the authors find. Decoding performance is similar between the two, and if anything, somewhat higher for the best kinematic variables. The new Supp. Fig. 9 suggests that some aspects of the neural responses are not captured by the kinematic variables (although that figure would be more informative if it also showed what "weakly-correlated" for the kinematic variables actually means in terms of a number). But an alternative reading of that figure would be that most PCs are substantially less well decodable in the periods of "weak-correlation", suggesting a weak encoding of tongue shape. So overall I am not sure that the data strongly supports the current (implied) statement that "sensorimotor cortex [encodes ...] detailed tongue shape" (line 184).

Thank you for this clear analysis. We agree with your interpretation of this figure, and do not want to overstate our results, so we dove deeper and performed an analysis where we plotted each shape variable's decoding performance as a function of its instantaneous correlation with the three reference kinematic variables (see revised Supp. Fig. 9). The new results demonstrate that while some shape variables' decoding performance may be attributed to correlation with kinematic variables (e.g., Shape PC1's decline from left to right for both subjects), there are multiple shape variables for both subjects whose decoding is relatively invariant to correlations with kinematics (e.g., Shape PC4 for Monkey Ry and Shape PC2 for Monkey Ye).

Considering the difficulty inherent in interpreting the behavioral relevance of the PCs, and in comparing them across monkeys, I would recommend to show Supp. Fig. 1 separately for the two monkeys. Any differences across monkeys in the exact ordering/functional relevance of individual PCs could lead to overall smaller "peak" correlation values between kinematic variables and PCs when averaging across monkeys.

We separated Supp Fig 1 by individual.